# Neural Spline Flows

**Conor Durkan**[*]  **Artur Bekasov**[*]  **Iain Murray**  **George Papamakarios**
School of Informatics, University of Edinburgh
{conor.durkan, artur.bekasov, i.murray, g.papamakarios}@ed.ac.uk

## Abstract

A normalizing flow models a complex probability density as an invertible transformation of a simple base density. Flows based on either coupling or autoregressive transforms both offer exact density evaluation and sampling, but rely on the parameterization of an easily invertible elementwise transformation, whose choice determines the flexibility of these models. Building upon recent work, we propose a fully-differentiable module based on monotonic rational-quadratic splines, which enhances the flexibility of both coupling and autoregressive transforms while retaining analytic invertibility. We demonstrate that neural spline flows improve density estimation, variational inference, and generative modeling of images.

## 1  Introduction

Models that can reason about the joint distribution of high-dimensional random variables are central to modern unsupervised machine learning. Explicit density evaluation is required in many statistical procedures, while synthesis of novel examples can enable agents to imagine and plan in an environment prior to choosing a action. In recent years, the variational autoencoder [VAE, 29, 48] and generative adversarial network [GAN, 15] have received particular attention in the generative-modeling community, and both are capable of sampling with a single forward pass of a neural network. However, these models do not offer exact density evaluation, and can be difficult to train. On the other hand, autoregressive density estimators [13, 50, 56, 58, 59, 60] can be trained by maximum likelihood, but sampling requires a sequential loop over the output dimensions.

Flow-based models present an alternative approach to the above methods, and in some cases provide both exact density evaluation and sampling in a single neural-network pass. A *normalizing flow* models data $\mathbf{x}$ as the output of an invertible, differentiable transformation $\mathbf{f}$ of noise $\mathbf{u}$:

$$\mathbf{x} = \mathbf{f}(\mathbf{u}) \quad \text{where} \quad \mathbf{u} \sim \pi(\mathbf{u}). \tag{1}$$

The probability density of $\mathbf{x}$ under the flow is obtained by a change of variables:

$$p(\mathbf{x}) = \pi\big(\mathbf{f}^{-1}(\mathbf{x})\big) \left| \det\!\left(\frac{\partial \mathbf{f}^{-1}}{\partial \mathbf{x}}\right) \right|. \tag{2}$$

Intuitively, the function $\mathbf{f}$ compresses and expands the density of the noise distribution $\pi(\mathbf{u})$, and this change is quantified by the determinant of the Jacobian of the transformation. The noise distribution $\pi(\mathbf{u})$ is typically chosen to be simple, such as a standard normal, whereas the transformation $\mathbf{f}$ and its inverse $\mathbf{f}^{-1}$ are often implemented by composing a series of invertible neural-network modules. Given a dataset $\mathcal{D} = \big\{\mathbf{x}^{(n)}\big\}_{n=1}^{N}$, the flow is trained by maximizing the total log likelihood $\sum_n \log p\big(\mathbf{x}^{(n)}\big)$ with respect to the parameters of the transformation $\mathbf{f}$. In recent years, normalizing flows have received widespread attention in the machine-learning literature, seeing successful use in density estimation [10, 43], variational inference [30, 36, 46, 57], image, audio and video generation [26, 28, 32, 45], likelihood-free inference [44], and learning maximum-entropy distributions [34].

---

[*]Equal contribution

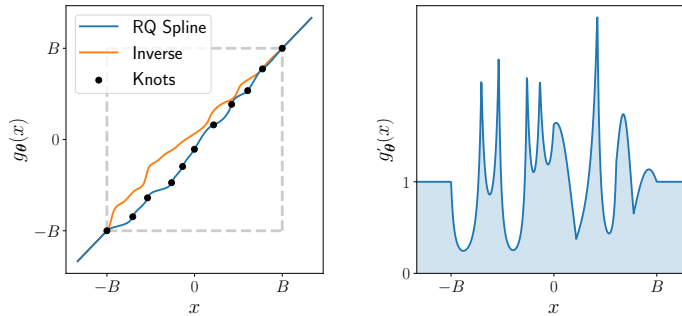

Figure 1: Monotonic rational-quadratic transforms are drop-in replacements for additive or affine transformations in coupling or autoregressive layers, greatly enhancing their flexibility while retaining exact invertibility. **Left**: A random monotonic rational-quadratic transform with $K = 10$ bins and linear tails is parameterized by a series of $K + 1$ 'knot' points in the plane, and the $K - 1$ derivatives at the internal knots. **Right**: Derivative of the transform on the left with respect to $x$. Monotonic rational-quadratic splines naturally induce multi-modality when used to transform random variables.

A flow is defined by specifying the bijective function $\mathbf{f}$ or its inverse $\mathbf{f}^{-1}$, usually with a neural network. Depending on the flow's intended use cases, there are practical constraints in addition to formal invertibility:

- To train a density estimator, we need to be able to evaluate the Jacobian determinant and the inverse function $\mathbf{f}^{-1}$ quickly. We don't evaluate $\mathbf{f}$, so the flow is usually defined by specifying $\mathbf{f}^{-1}$.
- If we wish to draw samples using eq. (1), we would like $\mathbf{f}$ to be available analytically, rather than having to invert $\mathbf{f}^{-1}$ with iterative or approximate methods.
- Ideally, we would like both $\mathbf{f}$ and $\mathbf{f}^{-1}$ to require only a single pass of a neural network to compute, so that both density evaluation and sampling can be performed quickly.

*Autoregressive flows* such as inverse autoregressive flow [IAF, 30] or masked autoregressive flow [MAF, 43] are $D$ times slower to invert than to evaluate, where $D$ is the dimensionality of $\mathbf{x}$. Subsequent work which enhances their flexibility has resulted in models which do not have an analytic inverse, and require numerical optimization to invert [22]. Flows based on *coupling layers* [NICE, RealNVP, 9, 10] have an analytic one-pass inverse, but are often less flexible than their autoregressive counterparts.

In this work, we propose a fully-differentiable module based on monotonic rational-quadratic splines which has an analytic inverse. The module acts as a drop-in replacement for the affine or additive transformations commonly found in coupling and autoregressive transforms. We demonstrate that this module significantly enhances the flexibility of both classes of flows, and in some cases brings the performance of coupling transforms on par with the best-known autoregressive flows. An illustration of our proposed transform is shown in fig. 1.

## 2 Background

### 2.1 Coupling transforms

A *coupling transform* $\phi$ [9] maps an input $\mathbf{x}$ to an output $\mathbf{y}$ in the following way:

1. Split the input $\mathbf{x}$ into two parts, $\mathbf{x} = [\mathbf{x}_{1:d-1}, \mathbf{x}_{d:D}]$.
2. Compute parameters $\boldsymbol{\theta} = \text{NN}(\mathbf{x}_{1:d-1})$, where NN is an arbitrary neural network.
3. Compute $y_i = g_{\boldsymbol{\theta}_i}(x_i)$ for $i = d, \ldots, D$ in parallel, where $g_{\boldsymbol{\theta}_i}$ is an invertible function parameterized by $\boldsymbol{\theta}_i$.
4. Set $\mathbf{y}_{1:d-1} = \mathbf{x}_{1:d-1}$, and return $\mathbf{y} = [\mathbf{y}_{1:d-1}, \mathbf{y}_{d:D}]$.

The Jacobian matrix of a coupling transform is lower triangular, since $\mathbf{y}_{d:D}$ is given by transforming $\mathbf{x}_{d:D}$ elementwise as a function of $\mathbf{x}_{1:d-1}$, and $\mathbf{y}_{1:d-1}$ is equal to $\mathbf{x}_{1:d-1}$. Thus, the Jacobian determinant of the coupling transform $\phi$ is given by $\det\left(\frac{\partial \phi}{\partial \mathbf{x}}\right) = \prod_{i=d}^{D} \frac{\partial g_{\boldsymbol{\theta}_i}}{\partial x_i}$, the product of the diagonal elements of the Jacobian.

Coupling transforms solve two important problems for normalizing flows: they have a tractable Jacobian determinant, and they can be inverted exactly in a single pass. The inverse of a coupling transform can be easily computed by running steps 1–4 above, this time inputting $\mathbf{y}$, and using $g_{\boldsymbol{\theta}_i}^{-1}$ to compute $\mathbf{x}_{d:D}$ in step 3. Multiple coupling layers can also be composed in a natural way to construct a normalizing flow with increased flexibility. A coupling transform can also be viewed as a special case of an autoregressive transform where we perform two splits of the input data instead of $D$, as noted by Papamakarios et al. [43]. In this way, advances in flows based on coupling transforms can be applied to autoregressive flows, and vice versa.

## 2.2 Invertible elementwise transformations

**Affine/additive**   Typically, the function $g_{\boldsymbol{\theta}_i}$ takes the form of an *additive* [9] or *affine* [10] transformation for computational ease. The affine transformation is given by:

$$g_{\boldsymbol{\theta}_i}(x_i) = \alpha_i x_i + \beta_i, \quad \text{where} \quad \boldsymbol{\theta}_i = \{\alpha_i, \beta_i\}, \tag{3}$$

and $\alpha_i$ is usually constrained to be positive. The additive transformation corresponds to the special case $\alpha_i = 1$. Both the affine and additive transformations are easy to invert, but they lack flexibility. Recalling that the base distribution of a flow is typically simple, flow-based models may struggle to model multi-modal or discontinuous densities using just affine or additive transformations, since they may find it difficult to compress and expand the density in a suitably nonlinear fashion (for an illustration, see appendix C.1). We aim to choose a more flexible $g_{\boldsymbol{\theta}_i}$, that is still differentiable and easy to invert.

**Polynomial splines**   Recently, Müller et al. [39] proposed a powerful generalization of the above affine transformations, based on monotonic piecewise polynomials. The idea is to restrict the input domain of $g_{\boldsymbol{\theta}_i}$ to the interval $[0, 1]$, partition the input domain into $K$ bins, and define $g_{\boldsymbol{\theta}_i}$ to be a simple polynomial segment within each bin. Müller et al. [39] restrict themselves to monotonically-increasing linear and quadratic polynomial segments, whose coefficients are parameterized by $\boldsymbol{\theta}_i$. Moreover, the polynomial segments are restricted to match at the bin boundaries so that $g_{\boldsymbol{\theta}_i}$ is continuous. Functions of this form, which interpolate between data using piecewise polynomials, are known as *polynomial splines*.

**Cubic splines**   In a previous iteration of this work [11], we explored the *cubic-spline flow*, a natural extension to the framework of Müller et al. [39]. We proposed to implement $g_{\boldsymbol{\theta}_i}$ as a *monotonic cubic spline* [54], where $g_{\boldsymbol{\theta}_i}$ is defined to be a monotonically-increasing cubic polynomial in each bin. By composing coupling layers featuring elementwise monotonic cubic-spline transforms with invertible linear transformations, we found flows of this type to be much more flexible than the standard coupling-layer models in the style of RealNVP [10], achieving similar results to autoregressive models on a suite of density-estimation tasks.

Like Müller et al. [39], our spline transform and its inverse were defined only on the interval $[0, 1]$. To ensure that the input is always between 0 and 1, we placed a sigmoid transformation before each coupling layer, and a logit transformation after each coupling layer. These transformations allow the spline transform to be composed with linear layers, which have an unconstrained domain. However, the limitations of 32-bit floating point precision mean that in practice the sigmoid saturates for inputs outside the approximate range of $[-13, 13]$, which results in numerical difficulties. In addition, computing the inverse of the transform requires inverting a cubic polynomial, which is prone to numerical instability if not carefully treated [1]. In section 3.1 we propose a modified method based on rational-quadratic splines which overcomes these difficulties.

## 2.3 Invertible linear transformations

To ensure all input variables can interact with each other, it is common to randomly permute the dimensions of intermediate layers in a normalizing flow. Permutation is an invertible linear transformation, with absolute determinant equal to 1. Oliva et al. [41] generalized permutations to a more general class of linear transformations, and Kingma and Dhariwal [28] demonstrated improvements on a range of image tasks. In particular, a linear transformation with matrix $\mathbf{W}$ is parameterized in terms of its *LU-decomposition* $\mathbf{W} = \mathbf{PLU}$, where $\mathbf{P}$ is a fixed permutation matrix, $\mathbf{L}$ is lower triangular with ones on the diagonal, and $\mathbf{U}$ is upper triangular. By restricting the diagonal elements of $\mathbf{U}$ to be positive, $\mathbf{W}$ is guaranteed to be invertible.

By making use of the LU-decomposition, both the determinant and the inverse of the linear transformation can be computed efficiently. First, the determinant of $\mathbf{W}$ can be calculated in $\mathcal{O}(D)$ time as the product of the diagonal elements of $\mathbf{U}$. Second, inverting the linear transformation can be done by solving two triangular systems, one for $\mathbf{U}$ and one for $\mathbf{L}$, each of which costs $\mathcal{O}(D^2 M)$ time where $M$ is the batch size. Alternatively, we can pay a one-time cost of $\mathcal{O}(D^3)$ to explicitly compute $\mathbf{W}^{-1}$, which can then be cached for re-use.

## 3 Method

### 3.1 Monotonic rational-quadratic transforms

We propose to implement the function $g_{\boldsymbol{\theta}_i}$ using *monotonic rational-quadratic splines* as a building block, where each bin is defined by a monotonically-increasing rational-quadratic function. A rational-quadratic function takes the form of a quotient of two quadratic polynomials. Rational-quadratic functions are easily differentiable, and since we consider only monotonic segments of these functions, they are also analytically invertible. Nevertheless, they are strictly more flexible than quadratic functions, and allow direct parameterization of the derivatives and heights at each knot. In our implementation, we use the method of Gregory and Delbourgo [17] to parameterize a monotonic rational-quadratic spline. The spline itself maps an interval $[-B, B]$ to $[-B, B]$. We define the transformation outside this range as the identity, resulting in linear 'tails', so that the overall transformation can take unconstrained inputs.

The spline uses $K$ different rational-quadratic functions, with boundaries set by $K+1$ coordinates $\{(x^{(k)}, y^{(k)})\}_{k=0}^{K}$ known as *knots*. The knots monotonically increase between $(x^{(0)}, y^{(0)}) = (-B, -B)$ and $(x^{(K)}, y^{(K)}) = (B, B)$. We give the spline $K-1$ arbitrary positive values $\{\delta^{(k)}\}_{k=1}^{K-1}$ for the derivatives at the internal points, and set the boundary derivatives $\delta^{(0)} = \delta^{(K)} = 1$ to match the linear tails. If the derivatives are not matched in this way, the transformation is still continuous, but its derivative can have jump discontinuities at the boundary points. This in turn makes the log-likelihood training objective discontinuous, which in our experience manifested itself in numerical issues and failed optimization.

The method constructs a monotonic, continuously-differentiable, rational-quadratic spline which passes through the knots, with the given derivatives at the knots. Defining $s_k = \left(y^{k+1} - y^k\right) / \left(x^{k+1} - x^k\right)$ and $\xi(x) = (x - x^k)/(x^{k+1} - x^k)$, the expression for the rational-quadratic $\alpha^{(k)}(\xi)/\beta^{(k)}(\xi)$ in the $k^{\text{th}}$ bin can be written

$$\frac{\alpha^{(k)}(\xi)}{\beta^{(k)}(\xi)} = y^{(k)} + \frac{\left(y^{(k+1)} - y^{(k)}\right)\left[s^{(k)}\xi^2 + \delta^{(k)}\xi(1 - \xi)\right]}{s^{(k)} + \left[\delta^{(k+1)} + \delta^{(k)} - 2s^{(k)}\right]\xi(1 - \xi)}. \tag{4}$$

Since the rational-quadratic transformation acts elementwise on an input vector and is monotonic, the logarithm of the absolute value of the determinant of its Jacobian can be computed as the sum of the logarithm of the derivatives of eq. (4) with respect to each of the transformed $x$ values in the input vector. It can be shown that

$$\frac{\mathrm{d}}{\mathrm{d}x}\left[\frac{\alpha^{(k)}(\xi)}{\beta^{(k)}(\xi)}\right] = \frac{\left(s^{(k)}\right)^2\left[\delta^{(k+1)}\xi^2 + 2s^{(k)}\xi(1 - \xi) + \delta^{(k)}(1 - \xi)^2\right]}{\left[s^{(k)} + \left[\delta^{(k+1)} + \delta^{(k)} - 2s^{(k)}\right]\xi(1 - \xi)\right]^2}. \tag{5}$$

Finally, the inverse of a rational-quadratic function can be computed analytically by inverting eq. (4), which amounts to solving for the roots of a quadratic equation. Because the transformation is monotonic, we can always determine which of the two quadratic roots is correct, and that the solution is given by $\xi(x) = 2c/\left(-b - \sqrt{b^2 - 4ac}\right)$, where

$$a = \left(y^{(k+1)} - y^{(k)}\right)\left[s^{(k)} - \delta^{(k)}\right] + \left(y - y^{(k)}\right)\left[\delta^{(k+1)} + \delta^{(k)} - 2s^{(k)}\right], \tag{6}$$

$$b = \left(y^{(k+1)} - y^{(k)}\right)\delta^{(k)} - \left(y - y^{(k)}\right)\left[\delta^{(k+1)} + \delta^{(k)} - 2s^{(k)}\right], \tag{7}$$

$$c = -s^{(k)}\left(y - y^{(k)}\right), \tag{8}$$

which can the be used to determine $x$. An instance of the rational-quadratic transform is illustrated in fig. 1, and appendix A.1 gives full details of the above expressions.

**Implementation**  The practical implementation of the monotonic rational-quadratic coupling transform is as follows:

1. A neural network NN takes $\mathbf{x}_{1:d-1}$ as input and outputs an unconstrained parameter vector $\boldsymbol{\theta}_i$ of length $3K-1$ for each $i = d, \ldots, D$.

2. Vector $\boldsymbol{\theta}_i$ is partitioned as $\boldsymbol{\theta}_i = \left[\boldsymbol{\theta}_i^w, \boldsymbol{\theta}_i^h, \boldsymbol{\theta}_i^d\right]$, where $\boldsymbol{\theta}_i^w$ and $\boldsymbol{\theta}_i^h$ have length $K$, and $\boldsymbol{\theta}_i^d$ has length $K-1$.

3. Vectors $\boldsymbol{\theta}_i^w$ and $\boldsymbol{\theta}_i^h$ are each passed through a softmax and multiplied by $2B$; the outputs are interpreted as the widths and heights of the $K$ bins, which must be positive and span the $[-B, B]$ interval. Cumulative sums of the $K$ bin widths and heights, each starting at $-B$, yield the $K+1$ knots $\left\{(x^{(k)}, y^{(k)})\right\}_{k=0}^{K}$.

4. The vector $\boldsymbol{\theta}_i^d$ is passed through a softplus function and is interpreted as the values of the derivatives $\left\{\delta^{(k)}\right\}_{k=1}^{K-1}$ at the internal knots.

Evaluating a rational-quadratic spline transform at location $x$ requires finding the bin in which $x$ lies, which can be done efficiently with binary search, since the bins are sorted. The Jacobian determinant can be computed in closed-form as a product of quotient derivatives, while the inverse requires solving a quadratic equation whose coefficients depend on the value to invert; we provide details of these procedures in appendix A.2 and appendix A.3. Unlike the additive and affine transformations, which have limited flexibility, a differentiable monotonic spline with sufficiently many bins can approximate any differentiable monotonic function on the specified interval $[-B, B]^2$, yet has a closed-form, tractable Jacobian determinant, and can be inverted analytically. Finally, our parameterization is fully-differentiable, which allows for training by gradient methods.

The above formulation can also easily be adapted for autoregressive transforms; each $\boldsymbol{\theta}_i$ can be computed as a function of $\mathbf{x}_{1:i-1}$ using an autoregressive neural network, and then all elements of $\mathbf{x}$ can be transformed at once. Inspired by this, we also introduce a set of splines for our coupling layers which act elementwise on $\mathbf{x}_{1:d-1}$ (the typically non-transformed variables), and whose parameters are optimized directly by stochastic gradient descent. This means that our coupling layer transforms all elements of $\mathbf{x}$ at once as follows:

$$\boldsymbol{\theta}_{1:d-1} = \text{Trainable parameters} \quad (9)$$
$$\boldsymbol{\theta}_{d:D} = \text{NN}(\mathbf{x}_{1:d-1}) \quad (10)$$
$$y_i = g_{\boldsymbol{\theta}_i}(x_i) \quad \text{for } i = 1, \ldots, D. \quad (11)$$

Training data    Flow density    Flow samples

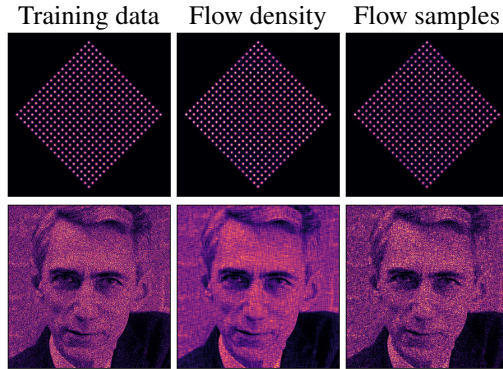

Figure 2 demonstrates the flexibility of our rational-quadratic coupling transform on synthetic two-dimensional datasets. Using just two coupling layers, each with $K = 128$ bins, the monotonic rational-quadratic spline transforms have no issue fitting complex, discontinuous densities with potentially hundreds of modes. In contrast, a coupling layer with affine transformations has significant difficulty with these tasks (see appendix C.1).

Figure 2: Qualitative results for two-dimensional synthetic datasets using RQ-NSF with two coupling layers.

## 3.2   Neural spline flows

The monotonic rational-quadratic spline transforms described in the previous section act as drop-in replacements for affine or additive transformations in both coupling and autoregressive transforms. When combined with alternating invertible linear transformations, we refer to the resulting class of normalizing flows as *rational-quadratic neural spline flows* (RQ-NSF), which may feature coupling layers, RQ-NSF (C), or autoregressive layers, RQ-NSF (AR). RQ-NSF (C) corresponds to Glow [28] with affine or additive transformations replaced with monotonic rational-quadratic transforms, where Glow itself is exactly RealNVP with permutations replaced by invertible linear transformations.

RQ-NSF (AR) corresponds to either IAF or MAF, depending on whether the flow parameterizes $\mathbf{f}$ or $\mathbf{f}^{-1}$, again with affine transformations replaced by monotonic rational-quadratic transforms, and also with permutations replaced with invertible linear layers. Overall, RQ-NSF resembles a traditional feed-forward neural network architecture, alternating between linear transformations and elementwise non-linearities, while retaining an exact, analytic inverse. In the case of RQ-NSF (C), the inverse is available in a single neural-network pass.

## 4    Related Work

**Invertible linear transformations**    Invertible linear transformations have found significant use in normalizing flows. Glow [28] replaces the permutation operation of RealNVP with an LU-decomposed linear transformation interpreted as a $1 \times 1$ convolution, yielding superior performance for image modeling. WaveGlow [45] and FloWaveNet [26] have also successfully adapted Glow for generative modeling of audio. Expanding on the invertible $1 \times 1$ convolution presented in Glow, Hoogeboom et al. [21] propose the *emerging convolution*, based on composing autoregressive convolutions in a manner analogous to an LU-decomposition, and the *periodic convolution*, which uses multiplication in the Fourier domain to perform convolution. Hoogeboom et al. [21] also introduce linear transformations based on the QR-decomposition, where the orthogonal matrix is parameterized by a sequence of Householder transformations [55].

**Invertible elementwise transformations**    Outside of those discussed in section 2.2, there has been much recent work in developing more flexible invertible elementwise transformations for normalizing flows. Flow++ [20] uses the CDF of a mixture of logistic distributions as a monotonic transformation in coupling layers, but requires bisection search to compute an inverse, since a closed form is not available. Non-linear squared flow [61] adds an inverse-quadratic perturbation to an affine transformation in an autoregressive flow, which is invertible under certain restrictions of the parameterization. Computing this inverse requires solving a cubic polynomial, and the overall transform is less flexible than a monotonic rational-quadratic spline. Sum-of-squares polynomial flow [SOS, 25] parameterizes a monotonic transformation by specifying the coefficients of a polynomial of some chosen degree which can be written as a sum of squares. For low-degree polynomials, an analytic inverse may be available, but the method would require an iterative solution in general.

Neural autoregressive flow [NAF, 22] replaces the affine transformation in MAF by parameterizing a monotonic neural network for each dimension. This greatly enhances the flexibility of the transformation, but the resulting model is again not analytically invertible. Block neural autoregressive flow [Block-NAF, 6] directly fits an autoregressive monotonic neural network end-to-end rather than parameterizing a sequence for each dimension as in NAF, but is also not analytically invertible.

**Continuous-time flows**    Rather than constructing a normalizing flow as a series of discrete steps, it is also possible to use a *continuous-time flow*, where the transformation from noise $\mathbf{u}$ to data $\mathbf{x}$ is described by an ordinary differential equation. Deep diffeomorphic flow [51] is one such instance, where the model is trained by backpropagation through an Euler integrator, and the Jacobian is computed approximately using a truncated power series and Hutchinson's trace estimator [23]. Neural ordinary differential equations [Neural ODEs, 3] define an additional ODE which describes the trajectory of the flow's gradient, avoiding the need to backpropagate through an ODE solver. A third ODE can be used to track the evolution of the log density, and the entire system can be solved with a suitable integrator. The resulting continuous-time flow is known as FFJORD [16]. Like flows based on coupling layers, FFJORD is also invertible in 'one pass', but here this term refers to solving a system of ODEs, rather than performing a single neural-network pass.

## 5    Experiments

In our experiments, the neural network NN which computes the parameters of the elementwise transformations is a residual network [18] with pre-activation residual blocks [19]. For autoregressive transformations, the layers must be masked so as to preserve autoregressive structure, and so we use the ResMADE architecture outlined by Nash and Durkan [40]. Preliminary results indicated only minor differences in setting the tail bound $B$ within the range $[1, 5]$, and so we fix a value $B = 3$ across experiments, and find this to work robustly. We also fix the number of bins $K = 8$ across

Table 1: Test log likelihood (in nats) for UCI datasets and BSDS300, with error bars corresponding to two standard deviations. FFJORD[†], NAF[†], Block-NAF[†], and SOS[†] report error bars across repeated runs rather than across the test set. Superscript[⋆] indicates results are taken from the existing literature. For validation results which can be used for comparison during model development, see table 6 in appendix B.1.

| Model | POWER | GAS | HEPMASS | MINIBOONE | BSDS300 |
|---|---|---|---|---|---|
| FFJORD[⋆†] | $0.46 \pm 0.01$ | $8.59 \pm 0.12$ | $-14.92 \pm 0.08$ | $-10.43 \pm 0.04$ | $157.40 \pm 0.19$ |
| GLOW | $0.42 \pm 0.01$ | $12.24 \pm 0.03$ | $-16.99 \pm 0.02$ | $-10.55 \pm 0.45$ | $156.95 \pm 0.28$ |
| Q-NSF (C) | $0.64 \pm 0.01$ | $12.80 \pm 0.02$ | $-15.35 \pm 0.02$ | $-9.35 \pm 0.44$ | $157.65 \pm 0.28$ |
| RQ-NSF (C) | $0.64 \pm 0.01$ | $13.09 \pm 0.02$ | $-14.75 \pm 0.03$ | $-9.67 \pm 0.47$ | $157.54 \pm 0.28$ |
| MAF | $0.45 \pm 0.01$ | $12.35 \pm 0.02$ | $-17.03 \pm 0.02$ | $-10.92 \pm 0.46$ | $156.95 \pm 0.28$ |
| Q-NSF (AR) | $0.66 \pm 0.01$ | $12.91 \pm 0.02$ | $-14.67 \pm 0.03$ | $-9.72 \pm 0.47$ | $157.42 \pm 0.28$ |
| NAF[⋆†] | $0.62 \pm 0.01$ | $11.96 \pm 0.33$ | $-15.09 \pm 0.40$ | $-8.86 \pm 0.15$ | $157.73 \pm 0.04$ |
| BLOCK-NAF[⋆†] | $0.61 \pm 0.01$ | $12.06 \pm 0.09$ | $-14.71 \pm 0.38$ | $-8.95 \pm 0.07$ | $157.36 \pm 0.03$ |
| SOS[⋆†] | $0.60 \pm 0.01$ | $11.99 \pm 0.41$ | $-15.15 \pm 0.10$ | $-8.90 \pm 0.11$ | $157.48 \pm 0.41$ |
| RQ-NSF (AR) | $0.66 \pm 0.01$ | $13.09 \pm 0.02$ | $-14.01 \pm 0.03$ | $-9.22 \pm 0.48$ | $157.31 \pm 0.28$ |

our experiments, unless otherwise noted. We implement all invertible linear transformations using the LU-decomposition, where the permutation matrix $\mathbf{P}$ is fixed at the beginning of training, and the product $\mathbf{LU}$ is initialized to the identity. For all non-image experiments, we define a flow 'step' as the composition of an invertible linear transformation with either a coupling or autoregressive transform, and we use 10 steps per flow in all our experiments, unless otherwise noted. All flows use a standard-normal noise distribution. We use the Adam optimizer [27], and anneal the learning rate according to a cosine schedule [35]. In some cases, we find applying dropout [53] in the residual blocks beneficial for regularization. Full experimental details are provided in appendix B. Code is available online at `https://github.com/bayesiains/nsf`.

## 5.1 Density estimation of tabular data

We first evaluate our proposed flows using a selection of datasets from the UCI machine-learning repository [7] and BSDS300 collection of natural images [38]. We follow the experimental setup and pre-processing of Papamakarios et al. [43], who make their data available online [42]. We also update their MAF results using our codebase with ResMADE and invertible linear layers instead of permutations, providing a stronger baseline. For comparison, we modify the quadratic splines of Müller et al. [39] to match the rational-quadratic transforms, by defining them on the range $[-B, B]$ instead of $[0, 1]$, and adding linear tails, also matching the boundary derivatives as in the rational-quadratic case. We denote this model Q-NSF. Our results are shown in table 1, where the mid-rule separates flows with one-pass inverse from autoregressive flows. We also include validation results for comparison during model development in table 6 in appendix B.1.

Both RQ-NSF (C) and RQ-NSF (AR) achieve state-of-the-art results for a normalizing flow on the Power, Gas, and Hepmass datasets, tied with Q-NSF (C) and Q-NSF (AR) on the Power dataset. Moreover, RQ-NSF (C) significantly outperforms both Glow and FFJORD, achieving scores competitive with the best autoregressive models. These results close the gap between autoregressive flows and flows based on coupling layers, and demonstrate that, in some cases, it may not be necessary to sacrifice one-pass sampling for density-estimation performance.

## 5.2 Improving the variational autoencoder

Next, we examine our proposed flows in the context of the variational autoencoder [VAE, 29, 48], where they can act as both flexible prior and approximate posterior distributions. For our experiments, we use dynamically binarized versions of the MNIST dataset of handwritten digits [33], and the EMNIST dataset variant featuring handwritten letters [5]. We measure the capacity of our flows to improve over the commonly used baseline of a standard-normal prior and diagonal-normal approximate posterior, as well as over either coupling or autoregressive distributions with affine transformations. Quantitative results are shown in table 2, and image samples in appendix C.

Table 2: Variational autoencoder test-set results (in nats) for the evidence lower bound (ELBO) and importance-weighted estimate of the log likelihood (computed as by Burda et al. [2] using 1000 importance samples). Error bars correspond to two standard deviations.

| POSTERIOR/PRIOR | MNIST | | EMNIST | |
|---|---|---|---|---|
| | ELBO | $\log p(\mathbf{x})$ | ELBO | $\log p(\mathbf{x})$ |
| BASELINE | $-85.61 \pm 0.51$ | $-81.31 \pm 0.43$ | $-125.89 \pm 0.41$ | $-120.88 \pm 0.38$ |
| GLOW | $-82.25 \pm 0.46$ | $-79.72 \pm 0.42$ | $-120.04 \pm 0.40$ | $-117.54 \pm 0.38$ |
| RQ-NSF (C) | $-82.08 \pm 0.46$ | $-79.63 \pm 0.42$ | $-119.74 \pm 0.40$ | $-117.35 \pm 0.38$ |
| IAF/MAF | $-82.56 \pm 0.48$ | $-79.95 \pm 0.43$ | $-119.85 \pm 0.40$ | $-117.47 \pm 0.38$ |
| RQ-NSF (AR) | $-82.14 \pm 0.47$ | $-79.71 \pm 0.43$ | $-119.49 \pm 0.40$ | $-117.28 \pm 0.38$ |

All models improve significantly over the baseline, but perform very similarly otherwise, with most featuring overlapping error bars. Considering the disparity in density-estimation performance in the previous section, this is likely due to flows with affine transformations being sufficient to model the latent space for these datasets, with little scope for RQ-NSF flows to demonstrate their increased flexibility. Nevertheless, it is worthwhile to highlight that RQ-NSF (C) is the first class of model which can potentially match the flexibility of autoregressive models, and which requires no modification for use as either a prior or approximate posterior, due to its one-pass invertibility.

### 5.3 Generative modeling of images

Finally, we evaluate neural spline flows as generative models of images, measuring their capacity to improve upon baseline models with affine transforms. In this section, we focus solely on flows with a one-pass inverse in the style of RealNVP [10] and Glow [28]. We use the CIFAR-10 [31] and downsampled $64 \times 64$ ImageNet [49, 60] datasets, with original 8-bit colour depth and with reduced 5-bit colour depth. We use Glow-like architectures with either affine (in the baseline model) or rational-quadratic coupling transforms, and provide full experimental detail in appendix B. Quantitative results are shown in table 3, and samples are shown in fig. 3 and appendix C.

RQ-NSF (C) improves upon the affine baseline in three out of four tasks, and the improvement is most significant on the 8-bit version of ImageNet64. At the same time, RQ-NSF (C) achieves scores that are competitive with the original Glow model, while significantly reducing the number of parameters required, in some cases by almost an order of magnitude. Figure 3 demonstrates that the model is capable of producing diverse, globally coherent samples which closely resemble real data. There is potential to further improve our results by replacing the uniform dequantization used in Glow with *variational dequantization*, and using more powerful networks with gating and self-attention mechanisms to parameterize the coupling transforms, both of which are explored by Ho et al. [20].

## 6 Discussion

Long-standing probabilistic models such as copulas [12] and Gaussianization [4] can simply represent complex marginal distributions that would require many layers of transformations in flow-based models like RealNVP and Glow. Differentiable spline-based coupling layers allow these flows, which are powerful ways to represent high-dimensional dependencies, to model distributions with complex shapes more quickly. Our results show that when we have enough data, the extra flexibility of spline-based layers leads to better generalization.

For tabular density estimation, both RQ-NSF (C) and RQ-NSF (AR) excel on Power, Gas, and Hepmass, the datasets with the highest ratio of data points to dimensionality from the five considered. In image experiments, RQ-NSF (C) achieves the best results on the ImageNet dataset, which has over an order of magnitude more data points than CIFAR-10. When the dimension is increased without a corresponding increase in dataset size, RQ-NSF still performs competitively with other approaches, but does not outperform them.

Overall, neural spline flows demonstrate that there is significant performance to be gained by upgrading the commonly-used affine transformations in coupling and autoregressive layers, without the need to sacrifice analytic invertibility. Monotonic spline transforms enable models based on coupling layers to achieve density-estimation performance on par with the best autoregressive flows,

Table 3: Test-set bits per dimension (BPD, lower is better) and parameter count for CIFAR-10 and ImageNet64 models. Superscript* indicates results are taken from the existing literature.

| Model | CIFAR-10 5-bit | | CIFAR-10 8-bit | | ImageNet64 5-bit | | ImageNet64 8-bit | |
|---|---|---|---|---|---|---|---|---|
| | BPD | Params | BPD | Params | BPD | Params | BPD | Params |
| Baseline | 1.70 | 5.2M | 3.41 | 11.1M | 1.81 | 14.3M | 3.91 | 14.3M |
| RQ-NSF (C) | 1.70 | 5.3M | 3.38 | 11.8M | 1.77 | 15.6M | 3.82 | 15.6M |
| Glow* | 1.67 | 44.0M | 3.35 | 44.0M | 1.76 | 110.9M | 3.81 | 110.9M |

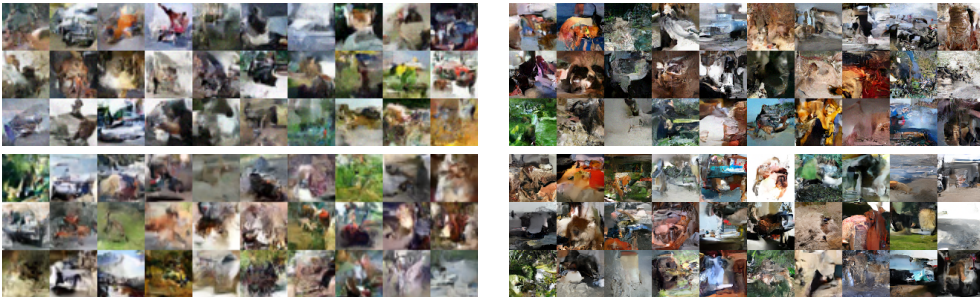

Figure 3: Samples from image models for 5-bit (top) and 8-bit (bottom) datasets. **Left:** CIFAR-10. **Right:** ImageNet64.

while retaining exact one-pass sampling. These models strike a novel middle ground between flexibility and practicality, providing a useful off-the-shelf tool for the enhancement of architectures like the variational autoencoder, while also improving parameter efficiency in generative modeling.

The proposed transforms scale to high-dimensional problems, as demonstrated empirically. The only non-constant operation added is the binning of the inputs according to the knot locations, which can be efficiently performed in $\mathcal{O}(\log_2 K)$ time for $K$ bins with binary search, since the knot locations are sorted. Moreover, due to the increased flexibility of the spline transforms, we find that we require fewer steps to build flexible flows, reducing the computational cost. In our experiments, which employ a linear $\mathcal{O}(K)$ search, we found rational-quadratic splines added approximately 30-40% to the wall-clock time for a single traning update compared to the same model with affine transformations. A potential drawback of the proposed method is a more involved implementation; we alleviate this by providing an extensive appendix with technical details, and a reference implementation in PyTorch. A third-party implementation has also been added to TensorFlow Probability [8].

Rational-quadratic transforms are also a useful differentiable and invertible module in their own right, which could be included in many models that can be trained end-to-end. For instance, monotonic warping functions with a tractable Jacobian determinant are useful for supervised learning [52]. More generally, invertibility can be useful for training very large networks, since activations can be recomputed on-the-fly for backpropagation, meaning gradient computation requires memory which is constant instead of linear in the depth of the network [14, 37]. Monotonic splines are one way of constructing invertible elementwise transformations, but there may be others. The benefits of research in this direction are clear, and so we look forward to future work in this area.

## Acknowledgements

This work was supported in part by the EPSRC Centre for Doctoral Training in Data Science, funded by the UK Engineering and Physical Sciences Research Council (grant EP/L016427/1) and the University of Edinburgh. George Papamakarios was also supported by Microsoft Research through its PhD Scholarship Programme.

## Footnotes

[2] By definition of the derivative, a differentiable monotonic function is locally linear everywhere, and can thus be approximated by a piecewise linear function arbitrarily well given sufficiently many bins. For a fixed and finite number of bins such universality does not hold, but this limit argument is similar in spirit to the universality proof of Huang et al. [22], and the universal approximation capabilities of neural networks in general.

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
