[Supplementary Material]

# Appendix for Neural Spline Flows

## A  Monotonic rational-quadratic transforms

### A.1  Parameterization of the spline

We here include an outline of the method of Gregory and Delbourgo [17] which closely matches the original paper. Let $\big\{ \big( x^{(k)}, y^{(k)} \big) \big\}_{k=0}^{K}$ be a given set of knot points in the plane which satisfy

$$(x^{(0)}, y^{(0)}) = (-B, -B), \tag{12}$$

$$(x^{(K)}, y^{(K)}) = (B, B), \tag{13}$$

$$x^{(k)} < x^{(k+1)} \text{ and } y^{(k)} < y^{(k+1)} \quad \text{for all } k = 0, \ldots, K-1. \tag{14}$$

Let $\big\{ \delta^{(k)} \big\}_{k=0}^{K}$ be the non-negative derivative values at these knot points (we take $\delta^{(0)} = \delta^{(K)} = 1$ so that the spline matches the derivative of the linear tails). Given these quantities, the algorithm of Gregory and Delbourgo [17] defines a monotonic rational-quadratic spline which passes through each knot and has the given derivative value at each knot as follows:

1. Let $w^{(k)} = x^{(k+1)} - x^{(k)}$ be the bin widths, and $s^{(k)} = \big( y^{(k+1)} - y^{(k)} \big)/w^{(k)}$ be the slopes of the lines joining the co-ordinates.

2. For $x \in \big[ x^{(k)}, x^{(k+1)} \big]$, let $\xi = \big( x - x^{(k)} \big)/w^{(k)}$, so that $\xi \in [0, 1]$.

3. Then, for $x \in \big[ x^{(k)}, x^{(k+1)} \big]$, $k = 0, \ldots, K-1$, define

$$g(x) = \frac{\alpha^{(k)}(\xi)}{\beta^{(k)}(\xi)}, \tag{15}$$

where

$$\alpha^{(k)}(\xi) = s^{(k)} y^{(k+1)} \xi^2 + \Big[ y^{(k)} \delta^{(k+1)} + y^{(k+1)} \delta^{(k)} \Big] \xi(1-\xi) + s^{(k)} y^{(k)} (1-\xi)^2, \tag{16}$$

$$\beta^{(k)}(\xi) = s^{(k)} \xi^2 + \Big[ \delta^{(k+1)} + \delta^{(k)} \Big] \xi(1-\xi) + s^{(k)} (1-\xi)^2. \tag{17}$$

Gregory and Delbourgo [17] note that $\beta^{(k)}(\xi)$ can be rewritten as

$$\beta^{(k)}(\xi) = s^{(k)} + \Big[ \delta^{(k+1)} + \delta^{(k)} - 2s^{(k)} \Big] \xi(1-\xi), \tag{18}$$

so that the quotient can be written as

$$\frac{\alpha^{(k)}(\xi)}{\beta^{(k)}(\xi)} = y^{(k)} + \frac{\big( y^{(k+1)} - y^{(k)} \big) \big[ s^{(k)} \xi^2 + \delta^{(k)} \xi(1-\xi) \big]}{s^{(k)} + \big[ \delta^{(k+1)} + \delta^{(k)} - 2s^{(k)} \big] \xi(1-\xi)}, \tag{19}$$

which is less prone to numerical issues, especially for small values of $s^{(k)}$. Gregory and Delbourgo [17] show that the spline defined by eq. (15) interpolates between the given knots, satisfies the derivative constraints at the knot points, and is monotonic on each bin. Rational-quadratic functions also provide flexibility over previous approaches: it is not possible to match arbitrary values and derivatives of a function at two boundary knots with a quadratic polynomial, or a monotonic segment of a cubic polynomial.

### A.2  Computing the derivative

The derivative of eq. (15) is given by the quotient rule:

$$\frac{\mathrm{d}}{\mathrm{d}x} \left[ \frac{\alpha^{(k)}(\xi)}{\beta^{(k)}(\xi)} \right] = \frac{\mathrm{d}}{\mathrm{d}\xi} \left[ \frac{\alpha^{(k)}(\xi)}{\beta^{(k)}(\xi)} \right] \frac{\mathrm{d}\xi}{\mathrm{d}x} = \frac{1}{w^{(k)}} \frac{\beta^{(k)}(\xi)\frac{\mathrm{d}}{\mathrm{d}\xi}\big[ \alpha^{(k)}(\xi) \big] - \alpha^{(k)}(\xi)\frac{\mathrm{d}}{\mathrm{d}\xi}\big[ \beta^{(k)}(\xi) \big]}{\big[ \beta^{(k)}(\xi) \big]^2}. \tag{20}$$

It can be shown that

$$\beta^{(k)}(\xi) \frac{\mathrm{d}\alpha^{(k)}(\xi)}{\mathrm{d}\xi} - \alpha^{(k)}(\xi) \frac{\mathrm{d}\beta^{(k)}(\xi)}{\mathrm{d}\xi} = w^{(k)} \big( s^{(k)} \big)^2 \Big[ \delta^{(k+1)} \xi^2 + 2s^{(k)} \xi(1-\xi) + \delta^{(k)} (1-\xi)^2 \Big], \tag{21}$$

so that

$$\frac{d}{dx}\left[\frac{\alpha^{(k)}(\xi)}{\beta^{(k)}(\xi)}\right] = \frac{\left(s^{(k)}\right)^2\left[\delta^{(k+1)}\xi^2 + 2s^{(k)}\xi(1-\xi) + \delta^{(k)}(1-\xi)^2\right]}{\left[s^{(k)} + \left[\delta^{(k+1)} + \delta^{(k)} - 2s^{(k)}\right]\xi(1-\xi)\right]^2}. \tag{22}$$

Since the rational-quadratic transform is monotonic and acts elementwise, the logarithm of the absolute value of the determinant of its Jacobian is given by a sum of the logarithm of eq. (22) for each transformed $x$.

### A.3 Computing the inverse

Computing the inverse of a monotonic rational-quadratic transformation when the value to invert lies in the tails is trivial. The problem of inversion is thus reduced to computing the inverse of the monotonic rational-quadratic spline. Consider a rational-quadratic function

$$y = \frac{\alpha(\xi(x))}{\beta(\xi(x))} = \frac{\alpha_0 + \alpha_1\xi(x) + \alpha_2\xi(x)^2}{\beta_0 + \beta_1\xi(x) + \beta_2\xi(x)^2}, \tag{23}$$

which arises as the result of the algorithm outlined in appendix A.1. The coefficients are such that the function is monotonically-increasing in its associated bin. Inverting the function involves solving a quadratic equation:

$$q(x) = \alpha(\xi(x)) - y\beta(\xi(x)) \tag{24}$$
$$= a\xi(x)^2 + b\xi(x) + c = 0, \tag{25}$$

where the coefficients depend on the target output $y$:

$$a = \alpha_2 - \beta_2 y, \quad b = \alpha_1 - \beta_1 y, \quad c = \alpha_0 - \beta_0 y. \tag{26}$$

Only one of the two solutions lies in the function's associated bin. To identify the solution in general, we identify that along the line of corresponding $(x, y)$ values, $q(x) = 0$ and so $\frac{dq}{dx} = 0$, where

$$\frac{dq}{dx} = \frac{\partial q}{\partial x} + \frac{\partial q}{\partial y}\frac{\partial y}{\partial x} \tag{27}$$

$$= \frac{\partial q}{\partial x} - \underbrace{\beta(\xi(x))}_{>0}\underbrace{\frac{\partial y}{\partial x}}_{>0} = 0. \tag{28}$$

We substituted a partial derivative of eq. (24), noted eq. (17) is positive, and noted that the spline $y(x)$ is monotonic and increasing. To satisfy eq. (28), $\frac{\partial q}{\partial x} > 0$, which corresponds to this solution to eq. (25):

$$\xi(x) = \frac{-b + \sqrt{b^2 - 4ac}}{2a} = \frac{2c}{-b - \sqrt{b^2 - 4ac}}, \tag{29}$$

where the first form is more commonly quoted, but the second form is numerically more precise when $4ac$ is small.

We can rearrange eq. (19) to show

$$a = \left(y^{(k+1)} - y^{(k)}\right)\left[s^{(k)} - \delta^{(k)}\right] + \left(y - y^{(k)}\right)\left[\delta^{(k+1)} + \delta^{(k)} - 2s^{(k)}\right], \tag{30}$$

$$b = \left(y^{(k+1)} - y^{(k)}\right)\delta^{(k)} - \left(y - y^{(k)}\right)\left[\delta^{(k+1)} + \delta^{(k)} - 2s^{(k)}\right], \tag{31}$$

$$c = -s^{(k)}\left(y - y^{(k)}\right), \tag{32}$$

which yields $\xi(x)$, which we can then use to determine the inverse $x$.

## B Experimental details

### B.1 Tabular density estimation

Model selection is performed using the standard validation splits for these datasets. We clip the norm of gradients to the range $[-5, 5]$, and find this helps stabilize training. We modify MAF by replacing

permutations with invertible linear layers. Hyperparameter settings are shown for coupling flows in table 4 and autoregressive flows in table 5. We include the dimensionality and number of training data points in each table for reference. For higher dimensional datasets such as Hepmass and BSDS300, we found increasing the number of coupling layers beneficial. This was not necessary for Miniboone, where overfitting was an issue due to the low number of data points.

Table 4: Hyperparameters for density-estimation results using coupling layers in section 5.1.

|  | POWER | GAS | HEPMASS | MINIBOONE | BSDS300 |
|---|---|---|---|---|---|
| DIMENSION | 6 | 8 | 21 | 43 | 63 |
| TRAIN DATA POINTS | 1,615,917 | 852,174 | 315,123 | 29,556 | 1,000,000 |
| BATCH SIZE | 512 | 512 | 256 | 128 | 512 |
| TRAINING STEPS | 400,000 | 400,000 | 400,000 | 200,000 | 400,000 |
| LEARNING RATE | 0.0005 | 0.0005 | 0.0005 | 0.0003 | 0.0005 |
| FLOW STEPS | 10 | 10 | 20 | 10 | 20 |
| RESIDUAL BLOCKS | 2 | 2 | 1 | 1 | 1 |
| HIDDEN FEATURES | 256 | 256 | 128 | 32 | 128 |
| BINS | 8 | 8 | 8 | 4 | 8 |
| DROPOUT | 0.0 | 0.1 | 0.2 | 0.2 | 0.2 |

Table 5: Hyperparameters for density-estimation results using autoregressive layers in section 5.1.

|  | POWER | GAS | HEPMASS | MINIBOONE | BSDS300 |
|---|---|---|---|---|---|
| DIMENSION | 6 | 8 | 21 | 43 | 63 |
| TRAIN DATA POINTS | 1,615,917 | 852,174 | 315,123 | 29,556 | 1,000,000 |
| BATCH SIZE | 512 | 512 | 512 | 64 | 512 |
| TRAINING STEPS | 400,000 | 400,000 | 400,000 | 250,000 | 400,000 |
| LEARNING RATE | 0.0005 | 0.0005 | 0.0005 | 0.0003 | 0.0005 |
| FLOW STEPS | 10 | 10 | 10 | 10 | 10 |
| RESIDUAL BLOCKS | 2 | 2 | 2 | 1 | 2 |
| HIDDEN FEATURES | 256 | 256 | 256 | 64 | 512 |
| BINS | 8 | 8 | 8 | 4 | 8 |
| DROPOUT | 0.0 | 0.1 | 0.2 | 0.2 | 0.2 |

Table 6: Validation log likelihood (in nats) for UCI datasets and BSDS300, with error bars corresponding to two standard deviations.

| MODEL | POWER | GAS | HEPMASS | MINIBOONE | BSDS300 |
|---|---|---|---|---|---|
| RQ-NSF (C) | $0.65 \pm 0.01$ | $13.08 \pm 0.02$ | $-14.75 \pm 0.06$ | $-9.03 \pm 0.43$ | $172.51 \pm 0.60$ |
| RQ-NSF (AR) | $0.67 \pm 0.01$ | $13.08 \pm 0.02$ | $-13.82 \pm 0.05$ | $-8.63 \pm 0.41$ | $172.5 \pm 0.59$ |

## B.2 Improving the variational autoencoder

We use the Adam optimizer [27] with default hyperparameters, annealing an initial learning rate of 0.0005 to 0 using a cosine schedule [35] over 150,000 training steps with batch size 256. We use a 'warm-up' phase for the KL divergence term of the loss, where the multiplier for this term is initialized to 0.5 and linearly increased to 1 over the first 10% of training. This modification initially reduces the penalty incurred by the approximate posterior in deviating from the prior, and similar schemes have been shown to improve VAE training dynamics [47]. Model selection is performed using a held-out validation set of 10,000 samples for MNIST, and 20,000 samples for EMNIST.

We use 32 latent features, and residual nets use 2 blocks, with 64 latent features for coupling layers, and 128 latent features for autoregressive layers. Both coupling and autoregressive flows use 10 steps. As with the tabular density-estimation experiments, we modify IAF [30] and MAF [43] by replacing permutations with invertible linear layers using an LU-decomposition. All NSF models use 8 bins. The encoder and decoder architectures are set up exactly as described by Nash and Durkan [40], and are similar to those used in IAF [30] and NAF [22].

Conditioning the approximate posterior distribution $q(\mathbf{z} \mid \mathbf{x})$ follows a multi-stage procedure. First, the encoder computes a context vector $\mathbf{h}$ of dimension $64$ as a function of the input $\mathbf{x}$. This vector is then mapped to the mean and diagonal covariance of a Gaussian distribution in the latent space. Then, $\mathbf{h}$ is also given as input to the residual nets in each of the flow's coupling or autoregressive layers, where it is concatenated with the input $\mathbf{z}$, mapped to the required number of hidden features, and also used to modulate the additive update of each residual block with a sigmoid gate. We found this scheme to work well across experiments.

### B.3 Generative modeling of images

For image-modeling experiments we use a Glow-like model architecture introduced by Kingma and Dhariwal [28, Section 3]. This involves stacking multiple steps for each level in the multi-scale architecture of Dinh et al. [10], where each step consists of an actnorm layer, an invertible $1 \times 1$ convolution and a coupling transform. For our RQ-NSF (C) model, we make the following modifications to the original Glow model: we replace affine coupling transforms with rational-quadratic coupling transforms, we go back to residual convolutional networks as used in RealNVP [10], and we use an additional $1 \times 1$ convolution at the end of each level of transforms. The basline model is the same as RQ-NSF (C), except that it uses affine coupling transforms instead of rational-quadratic ones. For CIFAR-10 experiments we do not factor out dimensions at the end of each level, but still use the squeezing operation to trade spatial resolution for depth.

For all experiments we use 3 residual blocks and batch normalization [24] in the residual networks which parameterize the coupling transforms. We use 7 steps per level for all experiments, resulting in a total of 21 coupling transforms for CIFAR-10, and 28 coupling transform for ImageNet64 (Glow models used by Kingma and Dhariwal [28] use 96 and 192 affine coupling transforms for CIFAR-10 and ImageNet64 respectively).

We use the Adam [27] optimizer with default $\beta_1$ and $\beta_2$ values. An initial learning rate of 0.0005 is annealed to 0 following a cosine schedule [35]. We train for 100,000 steps for 5-bit experiments, and for 200,000 steps for 8-bit experiments. To track the performance of our models, we split off 1% of the training data to use as a development set. Due to the resource requirements of the experiments, we perform a limited manual hyper-parameter exploration. Final values are reported in table 7.

We use a single NVIDIA Tesla P100 GPU card per CIFAR-10 experiment, and two such cards per ImageNet64 experiment. Training for 200,000 steps takes about 5 days with this setup.

Table 7: Hyperparameters for generative image-modeling results in section 5.3.

| DATASET | | BATCH SIZE | LEVELS | HIDDEN CHANNELS | BINS | DROPOUT |
|---|---|---|---|---|---|---|
| CIFAR-10 | 5-BIT | 512 | 3 | 64 | 2 | 0.2 |
| | 8-BIT | 512 | 3 | 96 | 4 | 0.2 |
| IMAGENET64 | 5-BIT | 256 | 4 | 96 | 8 | 0.1 |
| | 8-BIT | 256 | 4 | 96 | 8 | 0.0 |

## C   Additional experimental results

### C.1 Affine coupling transforms for 2D datasets

Densities fit by a model with two affine coupling layers on synthetic two-dimensional datasets are shown in fig. 4.

### C.2 Samples

Image samples for VAE experiments are shown in fig. 5. Additional samples for generative image-modeling experiments are shown in fig. 6.

Figure 4: Qualitative results for two-dimensional synthetic datasets using two affine coupling layers.

(a) MNIST

(b) EMNIST-letters

Figure 5: VAE samples. Top to bottom: training data, RQ-NSF (C), RQ-NSF (AR).

(a) CIFAR-10 5-bit

(b) ImageNet64 5-bit

(c) CIFAR-10 8-bit

(d) ImageNet64 8-bit

Figure 6: Additional image samples for generative image-modeling experiments.