[Reviews · NeurIPS 2019]

Reviewer 1



(Originality): The paper proposes a new elementwise transformation for use in coupling & autoregressive transforms. There have been multiple papers proposing such transformations recently. Notably, [Muller et al, 2018] proposes invertible transformations based on linear and quadratic splines. In an earlier iteration of this work, the authors proposed an extension of this in the form of cubic splines. This work extends on these by 1) allowing unconstrained inputs/outputs by letting the transformation be linear outside a range (improving numerical stability) and 2) using rational-quadratic splines instead of linear/quadratic/cubic splines (improving flexibility). This is clearly explained in the paper and related work is cited. (Quality): The paper is well executed. The paper proposes a new flexible elementwise transformation which can be used a replacement for the typical additive/affine transformations in coupling/autoregressive transformations, increasing their overall flexibility. This is demonstrated in the experiments where the authors show that by exchanging the additive/affine transformations in IAF/MAF/Glow by rational-quadratic transformations, they can 1) for some tabular datasets obtain state-of-the-art results, 2) show small but consistent improvements when used as a prior and variational distribution in a VAE and 3) obtain comparable results to Glow for images using 1/10th of the parameters. In the experiments on tabular data, the baselines are updated to be more similar to the proposed method. That is, MAF is updated to use ResMADE and invertible linear layers instead of permutations, while the quadratic spline flows of [Muller, 2018] are updated to have linear tails. This allows for a more fair comparison by isolating factors of improvement. (Clarity): The paper is generally well written and organized. The proposed method is clearly explained and supported by illustrative figures. While rational-quadratic splines were explained in detail in the appendix, I would have liked to see equations for the forward/inverse transformation and/or Jacobian determinant for the rational-quadratic transformation in the main text as this is a central point of the paper. Experimental details are given in the appendix, facilitating reproducibility of the results. Some comments/questions: - Was the coupling layer which transforms all variables used in all your experiments? And does this yield significant improvements over a standard coupling layer? - I would prefer to have boldface highlighting the top results within the C/AR part of the tables. (Significance): The proposed elementwise transformation adds a useful new module to the toolbox. As demonstrated in the experiments, this module can improve the flexibility of coupling and autoregressive transforms, in some cases leading to state-of-the-art results.

Reviewer 2



1) Originality The authors do a very good job at describing how their method fits within the family of building blocks of flow-based models. The literature review is, in my opinion, very thorough and well-written. 2) Quality The spline-based proposal of the authors for modelling makes a lot of sense, and the experimental results support show that using these rational-quadratic splines improves the flexibility of flows. A way of improving the paper would be to discuss more the computational cost implied by their method compared to simpler ones (e.g. affine transformations). 3) Clarity The review of flow-based models is extremely well-written (Sections 1 & 2). The final discussion is also very interesting. However, I think the authors should explain more clearly what monotonic rational-quadratic splines are (Section 3.1). Indeed, I think that understanding the paper without reading Appendix A is quite difficult, and some of the results form this Appendix could be moved in the main paper in future iterations. A few sentences are a bit vague: - l. 127: "Rational-quadratic functions are easily differentiable, more flexible than a polynomial in that they have an infinite Taylor-series expansion". There are families of functions with infinite Taylor-series expansions that are arguably not very flexible (like exp(-alpha*x)) - l. 136: "We [...] set the boundary derivatives to 1 to match the linear tails, which we found to be important for stable training." An experiment (for example in the appendix) would be nice to understand what you mean by "stable training". Also, do you have an explaination for this phenomenon? - l. 157: "Unlike the additive and affine transformations, which have limited flexibility, a monotonic spline with sufficiently many bins can approximate any continuous monotonic function on the specified interval" Could you add more details, and/or back the result by a citation? 4) Significance I'll repeat what I wrote in the contributions box: while the papers has essentially a single contribution, I think this is not a bad thing at all. Indeed the authors thorougly study (and, arguably, improve) a very specific and important part of flow-based models, which is valuable. It is an "incremental" paper in the good sense of the term. I would therefore recommend acceptance. --------- Post-rebuttal edit ------------- I've read your reviews and the rebuttal. I'm happy with the clarifications provided by the authors. This will be a nice contribution to the quickly growing field of flows.

Reviewer 3



The paper introduces a transformation which is elementwise strictly monotone (a bijection, i.e. invertible in principle) as well analytically invertible. The goal is to improve the flow's flexibility without losing access to an analytical inverse---which then enables *both* sampling and likelihood assessment. The technique is compatible both with couplings and autoregressive flows. The general idea is based on dividing a segment (say from -B to B) in bins by predicting bin widths (determined by ordered knots -- computed via cumulative sum of unordered predictions) and heights. Parameterising the transformation also involves predicting positive derivatives at the knots. This paper employs the parameterisation of Gregory and Delbourgo (1982). For the parts the paper does discuss, it does so reasonably clearly. But the parameterisation itself (key to flexibility) and the inverse transform (one of the big selling points) are considered supplementary and only explained in an Appendix. The empirical analysis shows a mixture of competitive results and state-of-the-art results, but generally never worse. In particular, density estimation shows clearer advantage for the proposed splines. The VAE setting was limited to MNIST and EMNIST -- it was unclear to me why not use all the typical benchmarks (Freyfaces, Omniglot, Caltech101), especially in light of a comment by the authors that suggest the datasets are too simple for their flow to shine. The authors also show results competitive with Glow in generative modelling (CIFAR and ImageNet) while using a rather compact model (up to 10x smaller parameter footprint). Whether or not the proposed flow will be widely used will largely depend on practical concerns regarding implementation (and availability of code): for example, it was unclear to me what difficulties (if any) underlie the parameterisation and the computation of the inverse.

[Author Response · NeurIPS 2019]

We would first like to thank the reviewers for their time and consideration.

Both **R2 and R3** wish us to elaborate on claims of **universality**. Firstly, we thank the reviewers for drawing our
attention to the imprecise statement on L156-159, which should read 'a *differentiable* monotonic spline with sufficiently
many bins can approximate any ~~continuous~~ *differentiable* monotonic function on the specified interval'. By definition
of the derivative, a differentiable monotonic function is locally linear everywhere, and can thus be approximated
by a piecewise linear function arbitrarily well given sufficiently many bins. Since rational-quadratic functions are
differentiable, they are also locally linear, and in the limit of number of bins, each segment becomes linear. Satisfying
differentiability everywhere means that continuous densities which are transformed by the spline do not become
discontinuous, so restricting ourselves to this class of monotonic functions is reasonable, and we also note that all
monotonic functions are differentiable almost everywhere regardless. Finally, as **R3** notes, for a fixed and finite number
of bins, this universality of course does not hold, but our limit argument is similar to that used both for universality of
neural networks in general, as well as that used by Neural Autoregressive Flows for their monotonic neural networks.

**R2** correctly notes that families of functions with **infinite Taylor-series expansion** may not necessarily be very flexible.
We will change this statement to highlight the precise flexibility that is added (ability to set derivatives and heights at
each knot location), which if achieved by increasing the polynomial degree would make it more difficult to accurately
invert the function.

**R2** asks about **fixing the boundary derivatives** of the splines to match the linear tails with slope 1. If the derivatives
at the boundaries of the splines are not set to match the values in the tails, the transformation is still continuous, but its
derivative can have jump discontinuities at the boundary points. Since the derivatives of the transformations are the
quantities which are accumulated by the change of variables to specify the exact log-likelihood, discontinuous derivatives
mean that the log-likelihood training objective becomes discontinuous, which in our experience manifested itself in
numerical issues and failed optimization. We observed this with both our proposed rational-quadratic transformations,
as well as when modifying the quadratic splines of Muller et al. to include linear tails.

Regarding the **VAE experiments**, **R3** asks why we have chosen tasks that do not seem to require flexible density models
of the latent space. Our intention when choosing the tasks was indeed to showcase the flexibility of the proposed flow;
to that end we picked challenging image datasets with a sufficient number of datapoints to be able to learn a complex
model of the latent space. Some of the typical benchmarks are only a fraction of the size, hence it is unlikely that
additional flexibility would be helpful. This is supported by results in FFJORD and Sylvester Normalizing Flows, where
flexible models do not always outperform simpler models on these datasets. We agree with **R3** that it is important to find
more challenging applications of VAEs that would truly test the latent-space modelling; finding such novel benchmarks
is outside of the scope of this work, but is an exciting research direction that would follow up on our findings.

**R1** asks about the **coupling layer which transforms all variables**. This addition provided marginal rather than
significant improvement, perhaps due to the fact that two successive coupling transforms which do not transform all
variables can perform the same function. Nevertheless, we used this technique in all coupling-layer models except for
the image experiments, where we found the marginal improvement not worth the extra parameter cost.

**All reviewers** mention the **importance of Appendix A**, and suggest some of its detail should be included in the main
text. We agree with the reviewers that this detail is essential for clear specification of our model, and we will move
some of this material to the main text.

Both **R2 and R3** ask for a discussion of **computational complexity and potential drawbacks** of the method. The
operations added by our proposed transforms do not depend on the input dimensionality, meaning the transforms scale
well to high-dimensional problems, as demonstrated empirically. The only non-constant operation is the binning of the
inputs according to the knot locations, which can be efficiently performed in $\mathcal{O}(\log_2 B)$ time for $B$ spline segments with
binary search (since the knot locations are sorted). Moreover, due to the increased flexibility of the spline transforms,
sequences of transforms required to build flexible flows are shorter (which we also observe empirically), reducing the
computational cost. We agree it is important to discuss these considerations in the paper, and intend to add a similar
discussion to the final version. One potential drawback of the proposed method is a more involved implementation; we
alleviate this by providing an extensive appendix with technical details, and a reference implementation in PyTorch.
Recently another group has independently reimplemented the transform in another framework using these resources.

**New results**: Finally, since submission we have extended our generative image-modelling experiments to include
baseline results for all rational-quadratic models, where we replace the spline transforms with affine transforms, keeping
all other choices fixed. Rational-quadratic splines improve upon the affine baseline in 3 out of 4 tasks, while matching
the baseline score on the 5-bit CIFAR-10 dataset. This provides evidence that the parameter-efficiency results that we
have observed in comparison with the full Glow model are not resulting from any changes unrelated to the elementwise
transforms, and that rational-quadratic splines do indeed increase the flexibility of the coupling transforms.

[Meta-Review · NeurIPS 2019]

This paper proposes a spline-based parametric approximation for monotonic functions as a module to be used in flow-based models. All reviewers were impressed with the work, which has exciting practical relevance across a variety of applications. Reviewers also appreciated the author response.